

# High spatial resolution CO₂ measurement using low-cost commercial sensors in Seoul megacity

JaeYoung Park[1], Jinho Ahn[1*], Jeongeun Kim[1], Nasrin Salehnia[1]

[1]School of Earth and Environment Sciences, Seoul National University, Seoul, 08826, Republic of Korea

*Correspondence to*: Jinho Ahn (jinhoahn@snu.ac.kr)

**Abstract.** Carbon dioxide ($CO_2$) is the most significant anthropogenic greenhouse gas. However, tracking $CO_2$ levels can be challenging due to the uneven distribution of concentrations and the high cost of sensors. In this study, we explored several correction techniques to enable the large-scale use of affordable $CO_2$ sensors, thereby enhancing the spatial resolution. We found that the low-cost $CO_2$ sensor (HT-2000) closely aligned with the trends observed in data from a more accurate sensor (LI-840a). By applying multiple-point linear regression, we reduced the root mean square error (RMSE) to only 1–2% of the measured value, which is accurate enough for urban monitoring at a local scale. Using a large network of low-cost sensors, we were able to map $CO_2$ concentration in detail, capture fine spatial variations, and gain a clearer understanding of emission patterns at an urban road intersection and within a tunnel.

## 1 Introduction

Carbon dioxide ($CO_2$) is the most significant anthropogenic greenhouse gas and the primary driver of the global climate crisis (IPCC, 2021). Currently, a larger proportion of the global population resides in urban areas than in rural regions. Urbanisation leads to disproportionate resource consumption in cities, contributing to approximately 70% of global $CO_2$ emissions linked to energy use (Rosenzweig et al., 2010; Gurney et al., 2020). Seoul is home to approximately 20% of the country's population and consumes approximately 9% of the nation's electricity (Korea Electric Power Corporation, 2023), despite occupying only

0.6% of the total land area (Seoul Metropolitan Government, 2024; Korean Ministry of Culture, Sports and Tourism, 2024). Monitoring urban $CO_2$ emissions is a critical step toward implementing effective reduction strategies and mitigating global warming. However, the highly diverse and heterogeneous land-use patterns in urban areas (Band et al., 2005; Olivo et al., 2017), combined with significant temporal variability in energy consumption (Olivo et al., 2017), result in substantial spatial and temporal variations in surface $CO_2$ concentrations (Park et al., 2022; Hong et al, 2023). This variability complicates

accurate estimation of $CO_2$ fluxes at small scales using a top-down approach, making urban flux calculations predominantly reliant on a bottom-up methods.

Previous studies have indicated that bottom-up emission estimates often suffer from considerable uncertainties. A comparison between downscaled global inventories frequently reveals differences exceeding 100% at the urban scale (Gately and Hutyra, 2017), and model results show substantial discrepancies when compared with bottom-up inventories (Gurney et al., 2019).

Ground-level $CO_2$ concentrations can be converted into atmospheric fluxes using flow measurement techniques such as eddy covariance (Burba et al., 2013; Vardag and Maiwald, 2023). Therefore, measuring ground-level $CO_2$ enables more accurate $CO_2$ flux estimation.

Numerous attempts have been made to measure urban $CO_2$ concentrations. One approach is satellite monitoring, which provides an accurate snapshot of $CO_2$ concentration at kilometre-scale spatial resolution (Kiel et al., 2021; Kort et al., 2012).

However, a major limitation of satellites is poor temporal resolution. All currently active greenhouse gas-monitoring satellites operate in low-Earth orbit, making continuous regional monitoring impossible. This limitation hinders the identification of diurnal patterns and obscures precise $CO_2$ sources. Furthermore, satellites measure total column $CO_2$, whose vertical profile varies significantly in the lower atmospheric (Roche et al., 2021), introducing additional inaccuracy. To achieve continuous temporal measurements, it is necessary to deploy gas-monitoring stations within cities (Imasu et al., 2018) or utilise mobile

platforms. Stationary monitoring stations offer superior temporal resolution and are useful for capturing daily, monthly, or seasonal patterns. However, budgetary constraints often limit spatial resolution, leading prior research efforts to rely on medium-precision, low-cost sensors (Spinelle et al., 2017; Arzoumanian et al., 2019).

This study presents an approach using a low-cost, pre-assembled $CO_2$ monitoring kit (including $CO_2$, relative humidity, and temperature sensors) to enable high-resolution $CO_2$ measurements in urban environments. We also present 2D visualisations

of ambient $CO_2$ levels measured in urban Seoul and within a tunnel in the city. Time-series data were corrected post hoc using a high-precision nondispersive infrared (NDIR) sensor, and we discuss methodologies for implementing effective correction schemes.

## 2 Materials and methods

### 2.1 Sensor description: HT-2000 and LI-840a

The HT-2000 is a commercially available device manufactured by Dongguan Xintai Instrument Co. Its primary component is a SenseAir S8 $CO_2$ sensor, which operates on the principle of the NDIR absorption. This sensor quantifies $CO_2$ concentrations by measuring infrared absorbance, following the Beer–Lambert law. The sensor has a measurement range of 400–2000 ppm, with an extended range of up to 10,000 ppm, albeit with reduced accuracy at higher concentrations (SenseAir, 2025). According to the manufacturer, its accuracy is within ±70 ppm or ±3% of the reading. Due to its relatively low-cost (HTi,

2023) compared with high-accuracy sensors used in similar applications, the HT-2000 provides a cost-effective solution for widespread deployment in laboratory settings.

In contrast, the LI-840a $CO_2$ analyser is a high-performance NDIR $CO_2$/$H_2O$ sensor manufactured by LI-COR Environmental a wide variety of applications. Unlike the HT-2000, which is an open-path sensor, the LI-840a operates as a closed-path system. However, both sensors are based on the same fundamental NDIR principles. The LI-840a boasts an accuracy better than 1.5%

of the reading value and an RMS noise level below 1 ppm. From our experience, the sensor's accuracy can be calibrated to





significantly surpass the manufacturer's specifications, often yielding an error of approximately 0.1 %. Despite its superior

performance, the high cost of the LI-840a limits the number of units that can be deployed simultaneously.

## 2.2 Site overview and measurement methods at Bongcheon Intersection

The Bongcheon Intersection is located directly above the Seoul National University subway station, which

serves approximately 40,000 passengers daily. Twenty locations in the vicinity of the station were selected for sensor

placement, representing the area's diverse land-use patterns. Among these, four points were located directly on the junction,

and two at nearby traffic crossings. Additionally, six points were situated near or on residential buildings, while the remaining

eight were near or on commercial buildings.

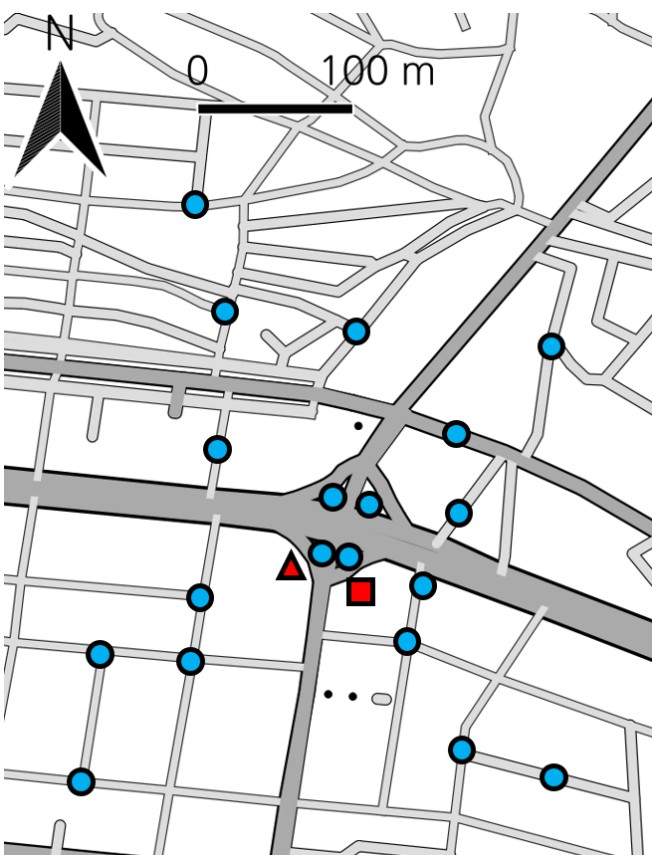

**Figure 1: Map of Bongcheon Intersection. Marked are correction data spots (red triangle [2022] and square [2023]) and HT-2000 measurement spots (blue circles for both dates). © OpenStreetMap Contributors 2024. Distributed under the Open Data Commons Open Database License (ODbL) v1.0.**

$CO_2$ levels around the Bongcheon Intersection were measured on 8 June 2022 and 8 February 2023 using 20 HT-2000 meters.

A pre-calibrated LI-840a analyser served as the reference instrument for calibrating the HT-2000 meters. The LI-840a was

calibrated using a two-point method with two different mixed gas canisters with known $CO_2$ concentrations (398 and 990 ppm).



We then corrected the 20 HT-2000 meters. The LI-840a was placed at a single location near the centre of the intersection using multiple-point linear regression.

Air was pumped through the LI-840a at a flow rate of 3 L/min using a MgClO₄ water vapour trap and aluminium tubing with a polymer coating, whereas the HT-2000 meters were exposed to open air. A 137-second moving average was applied to the

LI-840a data to calculate the time delay for each HT-2000 meter, ranging from 7 to 43 s. The smoothed LI-840a data were used to calibrate the vapour-corrected data from the HT-2000 meters (vapour pressure was removed using temperature, relative humidity, and average atmospheric pressure recorded by the HT-2000, as well as Tetens' approximation equation for saturation vapour pressure). This correction was performed using multiple-point linear regression. Finally, the HT-2000 meters were distributed and deployed at the Bongcheon Intersection for 1 h to collect data and create a detailed $CO_2$ concentration map.

Linear regression applied to each HT-2000 meter estimated the linear relationship between potentially multiple variables and a response. The model takes the following form:

$$y_i = \beta_0 + \beta_1 x_{i1} + \cdots + \beta_p x_{ip} + \varepsilon_i$$

where y represents the response variable, x represents the predictors, $\beta$ denotes the regression coefficients, and $\varepsilon$ is the error term. The subscript $i$ refers to the time point in the time-series regression. In this experiment, $y_i$ corresponds to the $CO_2$

concentration measured at time $i$ using the LI-840a analyser, while, $x_{i1}$, $x_{i2}$, and $x_{i3}$ represent the $CO_2$ concentration, temperature, and humidity, respectively. The model can be expressed in matrix form as follows:

$$\mathbf{y} = \mathbf{X}\boldsymbol{\beta} + \boldsymbol{\varepsilon}$$

where,

$$\mathbf{y} = \begin{bmatrix} y_1 \\ y_2 \\ \vdots \\ y_n \end{bmatrix}, \mathbf{X} = \begin{bmatrix} 1 & x_{11} & \cdots & x_{1p} \\ 1 & x_{12} & \cdots & x_{2p} \\ \vdots & \vdots & \ddots & \vdots \\ 1 & x_{n1} & \cdots & x_{np} \end{bmatrix}, \boldsymbol{\beta} = \begin{bmatrix} \beta_0 \\ \beta_1 \\ \beta_2 \\ \vdots \\ \beta_p \end{bmatrix}, \boldsymbol{\varepsilon} = \begin{bmatrix} \varepsilon_1 \\ \varepsilon_2 \\ \vdots \\ \varepsilon_n \end{bmatrix}$$

## 2.3 Site overview and measurement methods at Guryong Tunnel

The Guryong Tunnel, spanning 1,180 m, is located between the Gangnam and Seocho districts in Seoul. It experiences heavy traffic, with over 40 vehicles passing in a single direction each minute. $CO_2$ levels in the Guryong Tunnel were measured on 25 July 2024 and 21 November 2024, using 22 HT-2000 devices on 25 July and 21 devices on 21 November. To calibrate the results, all HT-2000 devices were placed in a sealed box the day before the measurements. The box was then flushed with

air containing known $CO_2$ concentrations (990 and 398 ppm) for 1 h to perform a two-point correction for each meter. The devices were powered overnight.

For the November 2024 experiments, an LI-840a device was used to calibrate the HT-2000 devices using the multi-point linear regression method, following the same procedure as at the Bongcheon Intersection. Correction data were obtained at the tunnel exit for 1 h between 09:30 and 10:30 AM. Subsequently, HT-2000 meters were placed inside the tunnel (Seoul-




bound direction) at semi-regular intervals, where $CO_2$ concentration were recorded for 1.5 h on 25 July (9:30 AM – 11:00 AM) and 20 h on 21 November (11:00 AM – 22 November 7:00 AM).

November data were corrected using two different methods: two-point and multi-point linear regressions. The results of these two methods were compared to assess their efficacy. The July measurement data were corrected using only a two-point method. Atmospheric pressure was not considered relevant because of its short measurement duration. Similarly, humidity and

temperature were excluded from the analysis because of high uncertainty in these measurements, and the potential improvement from including these factors was minimal compared with the associated errors.

The two-point regression method utilises two known reference points (high and low) and assumes that the sensor response is linear within this range. The sensor response was corrected using the following equation:

$$x_{sample} = x_{std1} + \frac{x_{std1} - x_{std2}}{y_{std1} - y_{std2}}\left(y_{sample} - y_{std1}\right)$$

where x denotes the actual value (or the assigned value for standard air); y represents the value displayed by the sensor; $x_{std1}$ and $x_{std2}$ are the known reference values for the high and low points, respectively; $y_{std1}$ and $y_{std2}$ are the corresponding sensor readings for the high and low points, respectively; and $y_{sample}$ is the sensor reading for the corrected sample.

## 3 Results

**3.1 Correction at Bongcheon Intersection**

Table 1 shows the average bias and time lag for each HT-2000 meter, along with the root mean square error (RMSE) values before and after correction. As indicated, the bias varied significantly across meters, with the worst-case deviation exceeding 200 ppm compared with the LI-840a measurements. Before linear correction, the median RMSE values were 42.07 ppm in 2022 and 30.60 ppm in 2023. After the correction, the median RMSE values decreased significantly to 5.10 ppm in 2022 and

4.09 ppm in 2023. Additionally, the RMSE for the $CO_2$ values compared with the LI-840a measurements (taken over the same period) is provided. RMSE was computed using the following equation:

$$RMSE = \sqrt{\frac{\sum_{i=1}^{n}(x_i^p - x_i^k)^2}{n}}$$

where $i$ is the time point; $x_i^p$ is the $CO_2$ value from the corrected HT-2000 sensor labelled p at $i$th time point; and $x_i^k$ is the $CO_2$ value from the LI-840a device at the same time point. The results demonstrate that multi-point linear regression

significantly the reduced the RMSE values by approximately one order-of-magnitude, from approximately 50 ppm (Figure 2c and 2d) to 5 ppm (Figure 2e and 2f).



**Table 1: Flat bias, time lag, RMSE (compared to LI-840a) for data pre- and post-correction from 2022 and 2023 Bongcheon Intersection measurements.**

| 2022 | | Root Mean Square values against LI-840a device | | Sensor labels | 2023 | | Root Mean Square values against LI-840a device | |
|---|---|---|---|---|---|---|---|---|
| Bias from LI-COR (ppm) | Time-lag (s) | Before correction (RMSE, ppm) | After correction (RMSE, ppm) | | Bias from LI-COR (ppm) | Time-lag (s) | Before correction (RMSE, ppm) | After correction (RMSE, ppm) |
| 138.07 | 39 | 138.85 | 5.15 | A | -19.64 | 17 | 19.73 | 4.02 |
| 227.12 | 37 | 227.80 | 4.85 | B | | | | |
| -2.75 | 37 | 8.26 | 4.57 | C | -31.10 | 27 | 31.11 | 3.92 |
| -36.14 | 43 | 36.59 | 5.36 | D | -4.83 | 15 | 7.13 | 4.21 |
| | | | | F | -50.71 | 27 | 50.61 | 4.56 |
| | | | | G | 25.17 | 22 | 26.56 | 5.15 |
| -43.61 | 37 | 43.73 | 5.13 | H | -23.09 | 7 | 23.13 | 4.12 |
| 205.55 | 40 | 206.26 | 5.30 | I | | | | |
| 172.97 | 34 | 173.65 | 4.38 | J | -15.33 | 16 | 15.47 | 3.72 |
| 62.60 | 38 | 63.71 | 5.54 | K | 5.63 | 21 | 7.54 | 3.93 |
| 58.29 | 41 | 59.43 | 4.98 | M | 39.85 | 13 | 40.57 | 3.97 |
| 18.30 | 35 | 20.35 | 4.71 | N | 148.23 | 14 | 148.71 | 3.95 |
| 40.79 | 36 | 42.07 | 4.83 | O | 84.70 | 23 | 85.28 | 4.42 |
| 12.35 | 34 | 15.05 | 5.20 | P | 36.76 | 16 | 37.44 | 3.83 |
| 76.77 | 44 | 77.80 | 4.94 | Q | 2.06 | 24 | 4.92 | 3.69 |
| -11.53 | 36 | 13.80 | 5.10 | R | 59.79 | 32 | 60.90 | 5.26 |
| -21.68 | 41 | 23.18 | 6.64 | S | -17.05 | 16 | 17.58 | 4.49 |
| 141.81 | 37 | 142.56 | 4.42 | T | | | | |
| 28.67 | 36 | 30.39 | 5.22 | U | -30.18 | 9 | 30.09 | 4.18 |
| | | | | V | 7.82 | 19 | 9.87 | 4.02 |
| | | | | W | -46.29 | 26 | 46.23 | 4.85 |
| 14.42 | 36 | 16.86 | 4.86 | X | 47.02 | 25 | 47.73 | 4.45 |
| -14.86 | 43 | 16.81 | 5.78 | Y | | | | |
| | | | | Z | 210.42 | 33 | 210.90 | 4.06 |







**Figure 2: CO₂ concentrations measured in Bongcheon Intersection, before and after correction. a) and b) Plots of CO₂ values measured by HT-2000 (grayscale) and LI-840a (red). c) and d) The LI-840a values smoothed with a 137-second window. Note that**





the HT-2000 sensor values closely follow the smoothed LI-840 values, though with fixed offsets. e) and f) Results after linear correction of the HT-2000 meters.

## 3.2 $CO_2$ concentration map at Bongcheon Intersection

The time-averaged (corrected) $CO_2$ values from the meters ranged from 348 to 482 ppm during the 2022 session, and from 457 to 512 ppm during the 2023 session. The $CO_2$ concentrations were interpolated using the *scatteredinterpolant* function in MATLAB. The results showed a distinct peak at the traffic intersection, with concentrations gradually decreasing as the distance from the intersection increased. In addition, the standard deviation of the $CO_2$ concentrations, which represents the variance, also exhibited a peak at the intersection.

Figure 3: Average and standard deviation of $CO_2$ concentration over time of Bongcheon Intersection. The top rows show average concentration of $CO_2$, the bottom rows show standard deviation. The left column shows the result for 2022 session; the right column shows the result for 2023 session. The standard deviation over time was calculated to show the variation in $CO_2$ concentration



**throughout the sessions. © OpenStreetMap Contributors 2024. Distributed under the Open Data Commons Open Database License (ODbL) v1.0.**

### 3.3 $CO_2$ concentration levels in the Guryong Tunnel

In the Guryong Tunnel, a clear $CO_2$ gradient was observed, with higher concentrations near the exit. While the entrance showed minimal temporal variation, the variability in $CO_2$ levels increased toward the exit. Additionally, $CO_2$ peaks occurred later at

positions closer to the exit (Figure 4), consistent with the piston effect induced by traffic movement (Chen et al., 1997). During the November measurements (post-correction), the highest recorded $CO_2$ concentration was 1,116 ppm, and the lowest was 421 ppm. Notable decreases in $CO_2$ levels were observed at approximately 12 PM and again at 8 PM (Figure 5).

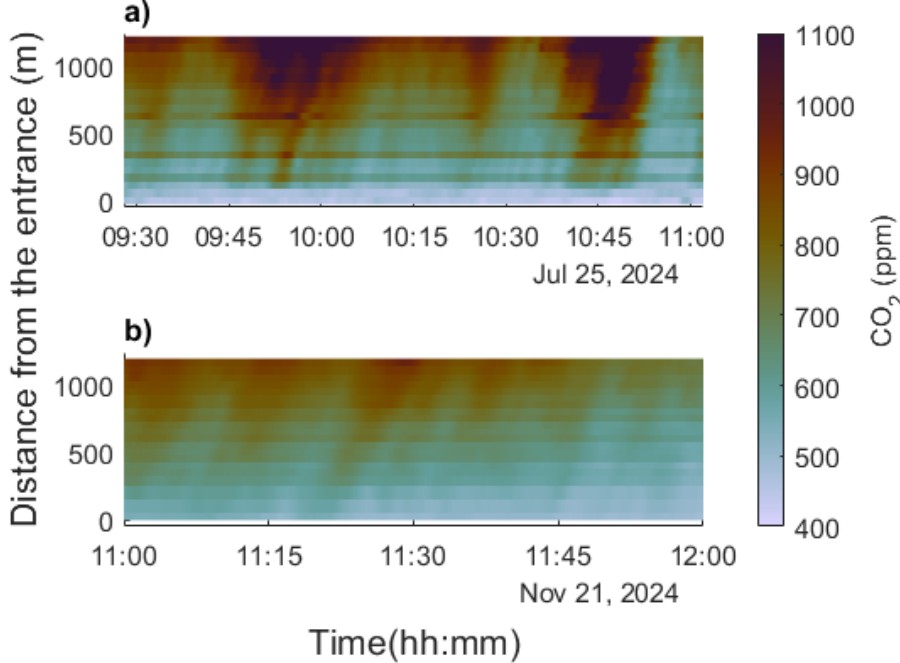

**Figure 4: $CO_2$ values measured (corrected) at the Guryong Tunnel, on July 25 (a) and November 21 (b, segment) in 2024.**





**Figure 5: Comparison of CO₂ values measured at Guryong Tunnel, corrected with different methods: raw data (a) vs 2-point regression (b) vs multi-point linear regression (c, d). Linear regression method shows significantly lower CO₂ concentration compared to 2-point calibration. Bottom image (d) is from multi-point linear regression attempt with time-lag compensation, compared to (c).**






**4 Discussion**

**4.1 Methodology**

**4.1.1 Comparison of calibration methods**

Initially, the error was assumed to be constant. Consequently, each HT-2000 device was treated as having a fixed bias relative to the actual $CO_2$ concentration for calibration. During calibration, the data from each HT-2000 unit were averaged and
compared with the average measured by the LI-840a. The HT-2000 data were then adjusted to match this average. For comparison, a multi-point linear regression method was used to calibrate the HT-2000 devices, assuming that the LI-840a readings were accurate. Both methods significantly reduced the RMSE values, with the multi-point linear regression method reducing the median RMSE by about 0.7 ppm, compared to the flat shift calibration method (Figure 6). This suggests that most of the errors stem from a constant offset from the true value. However, because multi-point linear regression also showed a
meaningful (~20%) reduction in the median RMSE, it is reasonable to conclude that scaling errors are also present in low-cost NDIR sensors.

Two calibration methods were applied in the Guryong Tunnel experiment: the two-point regression method and multi-point regression method. The two-point method resulted in a significant increase in the measured $CO_2$ concentration and a notable rise in the RMSE values, because the measured $CO_2$ concentration within the tunnel exceeded the concentrations of the gas
canisters used for the two-point regression. In contrast, the multi-point regression method led to a substantial decrease in RMSE (median RMSE 79.6 ppm before calibration vs. 12 ppm after calibration; Figure 7).



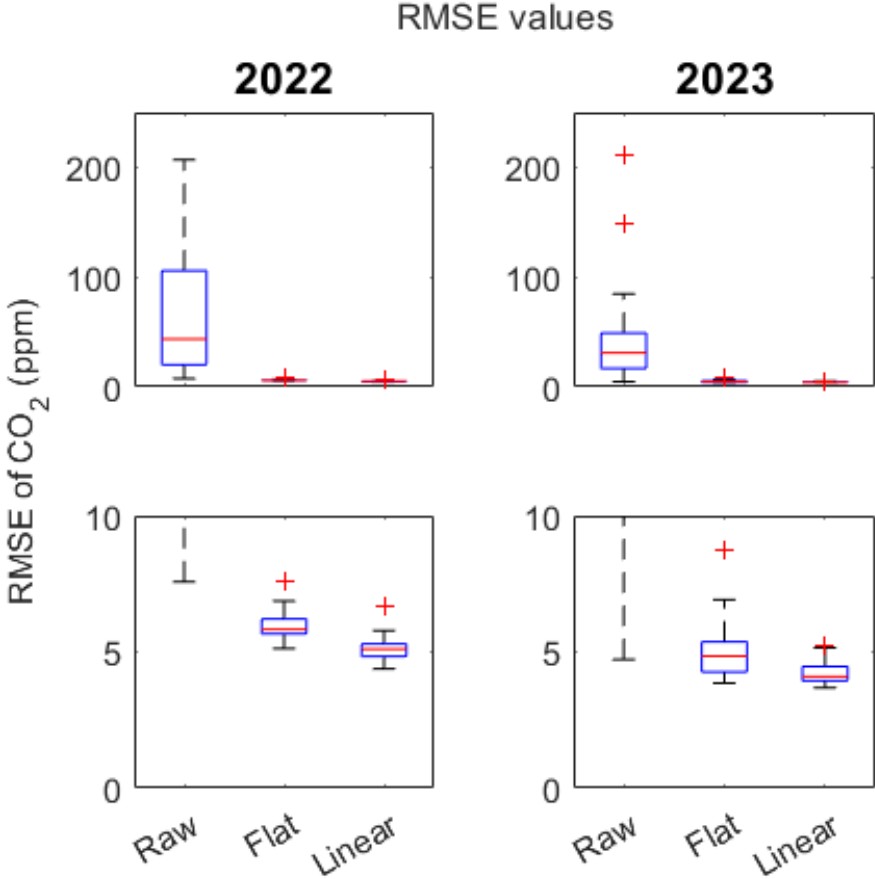

**Figure 6: Box diagram of RMSE values from $CO_2$ measurements made with HT-2000 at Bongcheon Intersection in 2022 and 2023 (against LI-840a). While both methods (flat shift and multi-point linear regression) reduced errors significantly, multi-point linear regression demonstrated a lower RMSE by about 0.7 ppm. The lower figures are enlarged sections of the top figures.**



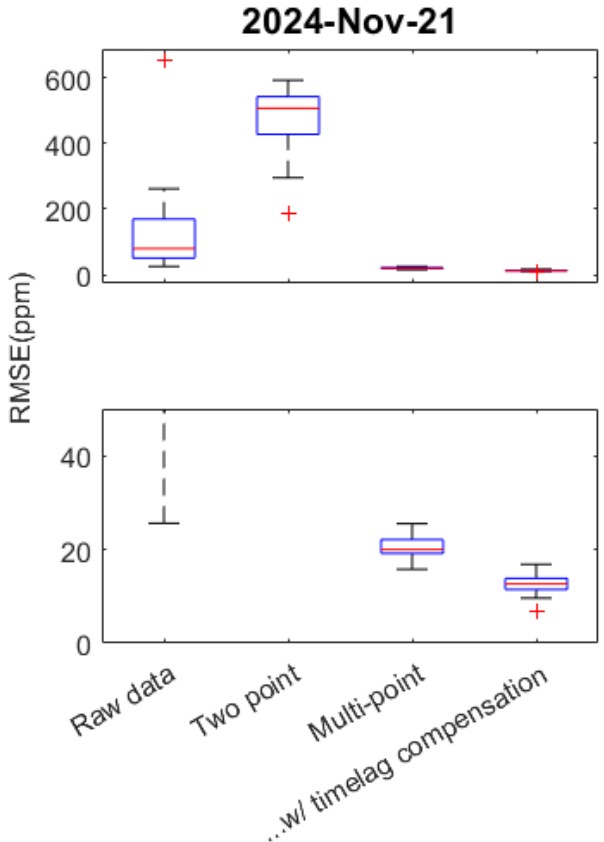

**Figure 7: Box diagram of RMSE of CO$_2$ concentrations measured in Guryong Tunnel with HT-2000 (against LI-840a). The HT-2000 meters were compared to results from a LI-840a device measuring from the same spot. The lower figure is an enlarged section of the top figure.**

### 4.1.2 Effect of time lag correction

For high temporal resolution measurements, synchronisation between devices is crucial; even minor timing discrepancies can lead to inaccurate representations of CO$_2$ movement. In this case, the time lag was calculated algorithmically using LI-840a as the ground truth, and the HT-2000 meters' time series were aligned accordingly. The effect of time-lag calibration varied between sessions. For the 2022 session, the average time lag of the HT-2000 devices (compared to LI-840a) was 38.58 s, and failing to apply time-lag correction resulted in a median RMSE increase of approximately 2 ppm. By contrast, for the 2023 session, the average time lag was 19.95 s, and not applying the correction led to a median RMSE increase of less than 0.5 ppm. The time delay of the sensors varied significantly between sessions, which can be explained by several factors. The first is the discrepancy between the internal clocks of the HT-2000 sensors and the laptop used to log HT-2000 data. However, because HT-2000 devices went through a synchronisation step before the measurements, such a large degree of variation does not seem plausible. Instead, we hypothesised that the MgClO$_4$ vapour trap affects the airflow within the airlines; airflow can vary





depending on whether the MgClO$_4$ grains are packed closely or loosely. The optimal smoothing windows for the 2022 and 2023 sessions were different (131 s for 2022, 232 s for 2023), which may also have been affected by the packing status of the
MgClO$_4$ grains.

### 4.1.3 Effect of temperature and relative humidity in multi-point linear regression

Absolute humidity and temperature were initially included as factors in the multi-point linear regression because we suspected residual interference in the low-cost NDIR sensor. To evaluate the impact of humidity and/or temperature in the regression, we performed a multi-point linear regression with different combinations of factors. Including absolute humidity or
temperature did not significantly change the median RMSE values, as shown in Figure 8 (ranging from 5.10 to 5.21 ppm in 2022 and from 4.09 to 4.71 ppm in 2023).



**Figure 8: Box diagram of RMSE values from CO$_2$ measurements made with HT-2000 at Bongcheon Intersection in 2022 and 2023. The x-axis indicates which variables were included in the linear regression.**

### 4.1.4 Evaluation of the feasibility of a common set of regression coefficients

Because the meters (and their internal sensors) are from the same company and model, it might seem reasonable to use a common set of regression coefficients for convenience and scientific consistency. However, the correction constants vary significantly, and using an average set of constants leads to significantly larger error values compared to individually corrected results. This finding aligns with previous studies (Martin et al., 2017), in which averaging the regression coefficients led to a
substantial increase in error (Figure 9).

In Martin et al. (2017), the experiment was conducted using Raspberry Pi-based modules, and extra effort was made to synchronize the sensors. The authors reported up to a 20-fold increase in RMSE values when a generalised set of coefficients was used to calibrate the sensors.



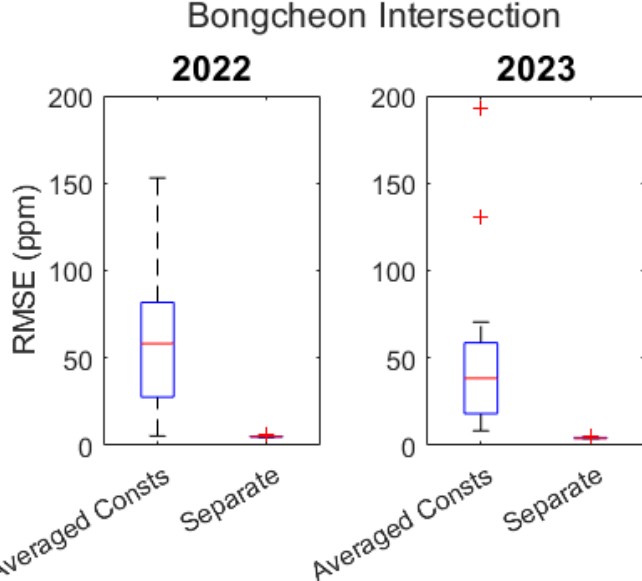


**Figure 9: Box diagram of RMSE values from CO$_2$ measurements made with HT-2000 at Bongcheon Intersection in 2022 and 2023, using separate constants vs. an averaged set of constants. Median values increased significantly when using averaged constants for correction.**

## 4.2 Main factors influencing CO$_2$ distribution at Bongcheon Intersection

As shown in Figure 3, traffic crossings exhibited the greatest variation and highest CO$_2$ concentrations. To analyse this further, videos were created to track CO$_2$ concentrations and their trends. In the CO$_2$ concentration videos (supplementary materials S1 and S3), small fluctuations were observed at the junction. Videos showing the rate of CO$_2$ concentration change (supplementary materials S2 and S4) revealed quasi-periodic variations at the junction. The four meters positioned at the junction were located on traffic islands, where the primary source of CO$_2$ is vehicular traffic. Specifically, vehicles idling at a

red light contribute to a concentration build-up at fixed locations, whereas vehicles in the right-turn lane do not have the same effect. Therefore, it is reasonable to associate these CO$_2$ changes with traffic signal cycles. A similar build-up was also observed in the northeastern corner of the 2023 video (supplementary material S4), where the responsible sensor was positioned near a traffic crossing. Additionally, we noted a local increase in CO$_2$ levels outside the intersection, likely caused by smaller, local traffic loads.

**4.3 Main factors influencing CO$_2$ distribution in the Guryong Tunnel**

The meters at the tunnel entrance showed slight temporal variation compared to those at the exit. Additionally, a noticeable trend emerged: both the CO$_2$ concentration and its variability increased as traffic moved through the tunnel. This observation



aligns with the previously described piston effect caused by traffic flow (Chen et al., 1997). Data from the 21 November measurements at the Guryong Tunnel revealed a sharp drop in $CO_2$ concentration after 8 PM. It was hypothesised that the $CO_2$

concentration approximately corresponds to the number of vehicles in the tunnel at a given time (vehicle density). To estimate this, the following equation was devised:

$$N = \frac{(Number\ of\ entering\ cars\ per\ unit\ time)}{(Traffic\ velocity)} \times (Tunnel\ length)$$

where $N$ is the vehicle density in the tunnel at a specific time. Although the number of vehicles entering the tunnel remained fairly constant around 8 PM, the average traffic velocity increased significantly, leading to a reduction in vehicle density. This

explains the rapid decrease in $CO_2$ concentration observed (Figure 10).





**Figure 10:** CO₂ concentration in the Guryong Tunnel, calculated number of vehicles inside the tunnel, and number of entering vehicles over time. (a) is identical to (d) in Figure 5; it represents the result of multi-point linear regression with time-lag correction. Total vehicle count (b) was calculated from interpolated number of vehicles entering the tunnel and hourly averaged traffic velocity. No significant drop in entering traffic (c) is observed around 8 PM, but average traffic speed increases significantly around at that time. The increased speed decreases the time a vehicle spends in the tunnel ultimately reducing the total number of vehicles inside.



## 5 Conclusions

This study demonstrates the feasibility of using low-cost NDIR sensors, such as the HT-2000, for high-resolution urban $CO_2$

monitoring when combined with effective correction techniques. By employing multi-point linear regression and time-lag correction, we significantly reduced the median RMSE of the HT-2000 sensors from over 10% to just 1–2% of the measured values, achieving an order-of-magnitude improvement in accuracy. This approach enables the deployment of large-scale sensor networks for detailed spatial and temporal monitoring of $CO_2$ concentrations in urban environments, where emissions exhibit significant variability owing to diverse land-use patterns and dynamic traffic conditions. The results highlight the importance

of individual sensor corrections, because the use of a common set of regression coefficients introduces substantial errors. Furthermore, this study underscores the influence of traffic dynamics on localised $CO_2$ distributions, as evidenced by the quasi-periodic variations at the Bongcheon Intersection and the piston effect observed in the Guryong Tunnel. These findings provide valuable insights for urban $CO_2$ flux estimation and underscore the potential of low-cost sensors to support targeted emission-reduction strategies. Future studies should focus on optimising correction methods and expanding sensor networks to capture

broader urban emission patterns, ultimately contributing to more accurate and actionable climate mitigation efforts.

*Author Contribution. Jinho Ahn designed the experiments, JaeYoung Park and JeongEun Kim carried them out. JaeYoung Park and Nasrin Salehnia developed code to analyze the results. JaeYoung Park prepared the manuscript with contributions from all co-authors.*


*Acknowledgements.* We would like to thank KwangJin Yim at the Center for Cryospheric Sciences, Seoul National University, for his dedicated assistance and support with air sample collection. We also thank Gangnam Roadway Management for the permission to enter the tunnel, and for the supplied traffic data. Finally, we would like to thank Editage (www.editage.co.kr) for English language editing. Maps were obtained from Openstreetmap (openstreetmap.org/copyright). The images are made

with MATLAB Version: 23.2.0.2391609 (R2023b) Update 2. The color schemes are from MATLAB and Crameri et al. (2020). This study is supported by National Research Foundation of Korea (2018R1A5A1024958; 2020H1D3A1A04081353; RS-2023-00291696; RS-2023-00278926) grant to JA.

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
