# Peer review of "High spatial resolution CO2 measurement using low-cost commercial sensors in Seoul megacity"

_EGUsphere, 2025_

## Referee Comment (RC1)

**General comments**

This paper presents results from a low-cost CO2 sensor deployment in Seoul. The authors deploy around two dozen low-cost sensors ("HT-2000" with SenseAir S8 Co2 sensors) along with a higher-quality sensor ("LI-840a" with LI-COR Environmental NDIR sensor) for calibration. The authors discuss different calibration methods and show that a multi-point linear regression that accounts for a time lag between sensors reduces RMSE the most, while temperature and humidity do not noticeably help reduce RMSE. After calibration, the authors show results from a 1-hour deployment at an intersection and a 20-hour deployment in a tunnel. The authors observe positive correlation between CO2 concentrations and the number of cars in the tunnel.

While the monitoring results presented in this paper are interesting and align with expected patterns (e.g., higher CO2 concentrations at the traffic intersection and during rush hour in the tunnel), it is not clear how long the calibration of the low-cost sensors would last during a longer-term deployment. The authors present only 1-hour of data from the first deployment at an intersection and 20-hours of data from the second deployment in a tunnel. I think the manuscript would benefit from additional analysis evaluating the robustness of the calibration procedure for longer deployments. For example, it is very important to know how often recalibration will be required. Furthermore, I think the organization of the paper could be improved. The authors present critical details about the calibration procedure in the Discussion section that I believe should be presented earlier in the methods section. I think it's important to fully understand the calibration procedure and efficacy before showing the CO2 monitoring results.

**Specific comments**
- Major concern: It is implied that the multiple linear regression calibration at the intersection is done at "correction" points where all low-cost and high-quality sensors are co-located based on the Figure 1 caption. However, this is not explicitly stated. Can the authors confirm that this is indeed the case, and if so, state it directly in the manuscript. If the sensors were not co-located before they were calibrated, then this could introduce possibly large errors due to spatial variability in CO2 concentrations.
- Major concern: It is implied that the data in Figure 2 is the data used to calibrate the low-cost sensors while they were co-located with the high-quality sensor, and that the spatial maps in Figure 3 are based on the low-cost measurements at the sensor locations shown in Figure 1. Can the authors confirm if this is the case? And if so, please state this directly. It would be useful to show not just the data used to calibrate (what I believe is Figure 2) but also the time series data used to generate the spatial maps. It is a bit confusing as currently presented, as the time series from each sensor in Figure 2 look nearly identical after calibration, but the spatial maps show clear spatial patterns.

- Major concern: since the calibration and deployment periods were relatively short, it is not clear if the low-cost sensors will continue to provide accurate measurements over longer deployment periods. Do the authors believe that this sensor network can be used for longer-term monitoring? If yes, how often will the low-cost sensors require re-calibration?
- Introduction: I think some discussion of other studies that perform urban-scale CO2 monitoring using low-cost sensors is missing. How is this study different? E.g., are there new bias-correction methods that result in more accurate measurements from the low-cost sensors? Is this the first time this type of monitoring is being performed in Seoul? Or is it the first time these specific sensors have been used for continuous monitoring?
- Section 2.1: It would be interesting, if possible, to list sensor prices. Perhaps just an order of magnitude. This would provide context and motivation for why the low-cost networks are important.
- L79: How was the length of the 137s moving time window selected? How sensitive is the calibration to this length?
- L80: How was the time delay calculated for each HT-2000 sensor? In general, more detail is needed on the calibration procedure in this section.
- L84: The authors point to low-cost sensors for better temporal coverage, but they only measure for 1 hour. Would it have been possible to leave the sensors deployed for longer?
- Section 2: the discussion of sensor calibration is fragmented and hard to follow. I recommend having a subsection devoted solely to the detailed discussion of sensor calibration methods. There is also some calibration information in Section 4.1 that I think belons in Section 2 before presenting results.
- Section 2: more discussion is needed about why different calibration methods and meteorological variables were used between the two study areas. Why were temperature and humidity excluded from the tunnel study, but included in the intersection study? Why was two-point calibration and multiple linear regression used at the tunnel, but only multiple linear regression at the intersection? After reading the whole article – I think the calibration results in section 4.1 belong in section 2 (before presenting the co2 results).
- Section 2.3: This section would really benefit from a map, similar to what was provided in section 2.2
- Section 3.1: Please list the coefficient values for the multiple linear regression, either in the main text or the SI. It's not clear how much of the correction is currently coming from the meteorological variables. After reading the whole article I see that Section 4.1.3 has some information on this. See earlier note about ordering. I still think it would be useful to list the coefficient values.
- I think Section 4.1 belongs in the methods section, not the discussion section. Several of my questions were answered when reading this section, and I would have liked to see calibration details before looking at the results.

**Technical corrections**

- L30: This sentence seems a bit out of place, since the authors did not conduct eddy covariance measurements. Instead, I think the authors should motivate by referencing other studies that use low-cost sensor networks to monitor emissions.
- L59: "The LI-840a boasts an accuracy better than 1.5% of the reading value and an RMS noise level below 1 ppm." I think this sentence should be stated the same way as the sentence describing the HT-2000 sensor to make for an easier comparison. Something like: "according to the manufacturer, its accuracy is within +/- XX ppm or +/- YY% of the reading." It's not clear if the RMS noise level of < 1ppm stated here is the same metric as the +/- 70ppm stated for the HT-2000.
- L60-61: This claim seems unsupported by the work in this manuscript. Is there another study you can reference to support the 0.1% accuracy claim?
- Fig 1: I only see 19 blue dots. Where is the 20th sensor located?
- L76: This sentence doesn't make sense: "The LI-840a was placed at a single location near the centre of the intersection using multiple-point linear regression." Perhaps this paragraph is out of order?
- L90: Clarify that the covariates in the multiple linear regression are the fields measured by the HT-2000 sensor.
- L142: Please state which interpolation method you are using with "*scatteredinterpolant.*" It's also worth noting that this method does not leverage any information about, e.g., atmospheric transport or the presence of buildings that would block the flow of CO2 between potential sources and the sensor locations.
- Figure 4: It's unclear what the "distance from the entrance" means. Is this being measured into the tunnel, with 0m at the entrance? If so, at what distance is the exit point? It would be helpful to mark this distance on the figure. Also see earlier note about providing a map of the tunnel experiment. How many sensors were there and how far apart were they within the tunnel?
- Figure 4: Needs some discussion of how the point data was spatial interpolated. Is it the same method as the intersection case study?
- Fig 10: please state where the traffic data are coming from

---

## Author Comment (AC1)

*We would like to thank the two anonymous reviewers and the Editor for their careful reviews of our manuscript. Below we present our point-by-point responses to all of the comments. The original comments from the reviewers are shown in black, while our responses are presented in blue.*

**Reviewer #1**

**General comments**

This paper presents results from a low-cost CO2 sensor deployment in Seoul. The authors deploy around two dozen low-cost sensors ("HT-2000" with SenseAir S8 Co2 sensors) along with a higher-quality sensor ("LI-840a" with LI-COR Environmental NDIR sensor) for calibration. The authors discuss different calibration methods and show that a multi-point linear regression that accounts for a time lag between sensors reduces RMSE the most, while temperature and humidity do not noticeably help reduce RMSE. After calibration, the authors show results from a 1-hour deployment at an intersection and a 20-hour deployment in a tunnel. The authors observe positive correlation between CO2 concentrations and the number of cars in the tunnel.

While the monitoring results presented in this paper are interesting and align with expected patterns (e.g., higher CO2 concentrations at the traffic intersection and during rush hour in the tunnel), it is not clear how long the calibration of the low-cost sensors would last during a longer-term deployment. The authors present only 1-hour of data from the first deployment at an intersection and 20-hours of data from the second deployment in a tunnel. I think the manuscript would benefit from additional analysis evaluating the robustness of the calibration procedure for longer deployments. For example, it is very important to know how often recalibration will be required. Furthermore, I think the organization of the paper could be improved. The authors present critical details about the calibration procedure in the Discussion section that I believe should be presented earlier in the methods section. I think it's important to fully understand the calibration procedure and efficacy before showing the CO2 monitoring results.

**Specific comments**

- Major concern: It is implied that the multiple linear regression calibration at the intersection is done at "correction" points where all low-cost and high-quality sensors are co-located based on the Figure 1 caption. However, this is not explicitly stated. Can the authors confirm that this is indeed the case, and if so, state it directly in the manuscript. If the sensors were not co-located before they were calibrated, then this could introduce possibly large errors due to spatial variability in CO2 concentrations.

  - The sensors were colocated to obtain the data for correction; now we have explicitly mentioned it in the manuscript and figure S2.

- Major concern: It is implied that the data in Figure 2 is the data used to calibrate the low-cost sensors while they were co-located with the high-quality sensor, and that the spatial maps in Figure 3 are based on the low-cost measurements at the sensor locations shown in Figure 1. Can the authors confirm if this is the case? And if so, please state this directly. It would be useful to show not just the data used to calibrate (what I believe is Figure 2) but also the time series data used to generate the spatial maps. It is a bit confusing as currently presented, as the time series from each sensor in Figure 2 look nearly identical after calibration, but the spatial maps show clear spatial patterns.

  - Figure S2 has been newly added to show the time series used for when low-cost sensors and LICOR were co-located for correction (i), (in transit (ii)), and HT-2000 sensors were deployed for measuring data to create spatial maps (iii).

[Figure]

**Figure S2: CO$_2$ concentrations measured in Bongcheon Intersection, before and after correction. a) and b) Plots of CO$_2$ values measured by HT-2000 (grayscale) and LI-840a (red), before any correction was applied. c) and d) The LI-840a values smoothed with a 137-second window. Note that the HT-2000 sensor values closely follow the smoothed LI-840 values, though with fixed offsets. e) and f) Results after linear correction of the HT-2000 meters. i)-iii) sections (labelled in a)) each represents: co-located measurement for obtaining data for correction of HT-2000 (i), moving HT-2000 sensors to its assigned location (ii), HT-2000 measurement in its assigned location (iii).**

- Major concern: since the calibration and deployment periods were relatively short, it is not clear if the low-cost sensors will continue to provide accurate

measurements over longer deployment periods. Do the authors believe that this sensor network can be used for longer-term monitoring? If yes, how often will the low-cost sensors require re-calibration?

- During measurements at the Guryong Tunnel, we found that the sensors remained stable for 24 hours. We expect the sensors to remain stable for a longer period, but the HT-2000's battery life is insufficient to operate them for more than 48 hours. We note this limitation in Section 2.1.

- Introduction: I think some discussion of other studies that perform urban-scale CO2 monitoring using low-cost sensors is missing. How is this study different? E.g., are there new bias-correction methods that result in more accurate measurements from the low-cost sensors? Is this the first time this type of monitoring is being performed in Seoul? Or is it the first time these specific sensors have been used for continuous monitoring?

  - We used very low-cost kits (85 USD for each) compared with other low-cost kits (~1000s USD). This enables a large number of sensors to be used at the same time to increase spatial resolution. To clarify, we added the following words in the main text:

    "This study presents an approach using a very low-cost, pre-assembled $CO_2$ monitoring kit (~85 USD; including $CO_2$, relative humidity, and temperature sensors) to enable high-resolution $CO_2$ measurements in urban environments."

- Section 2.1: It would be interesting, if possible, to list sensor prices. Perhaps just an order of magnitude. This would provide context and motivation for why the low-cost networks are important.

  - We added the sensor price of 85 USD in the manuscript.

- L79: How was the length of the 137s moving time window selected? How sensitive is the calibration to this length?

- The smoothing window of 137 seconds is chosen using an algorithm; that is, we have tried various lengths of smoothing window using a FOR loop for both correction periods and found out that 137 seconds yielded the least RMSE difference between the LI-COR and HT-2000 instruments. To clarify, we added the following words:

    "We have tried various lengths of smoothing window using a FOR loop for both correction periods and found out that 137 seconds yielded the least RMSE difference between the LI-COR and HT-2000 instruments."

- L80: How was the time delay calculated for each HT-2000 sensor? In general, more detail is needed on the calibration procedure in this section.

    - The time delay was also calculated algorithmically, by finding the "best match" between the LI-COR and HT-200. i.e., by running a FOR loop to vary the time delay, and calculating the standard deviation between the difference of HT-2000 and LI-840a. We added the following words:

        - "The time delay for each HT-200 was also calculated algorithmically, by finding the "best match" between the LI-COR and HT-200."

- L84: The authors point to low-cost sensors for better temporal coverage, but they only measure for 1 hour. Would it have been possible to leave the sensors deployed for longer?

    - During measurements at the Guryong Tunnel, we found that the sensors remained stable for 24 hours. We expect the sensors to remain stable for a longer period, but the HT-2000's battery life is insufficient to operate them for more than 48 hours. We note this limitation in Section 2.1.

- Section 2: the discussion of sensor calibration is fragmented and hard to follow. I recommend having a subsection devoted solely to the detailed discussion of sensor calibration methods. There is also some calibration information in Section 4.1 that I think belons in Section 2 before presenting results.

    - We added new Sections 2.3.3 through 2.3.6 and moved the content from the former Section 4.1 into them.

- Section 2: more discussion is needed about why different calibration methods and meteorological variables were used between the two study areas. Why were temperature and humidity excluded from the tunnel study, but included in the intersection study? Why was two-point calibration and multiple linear regression used at the tunnel, but only multiple linear regression at the intersection? After reading the whole article – I think the calibration results in section 4.1 belong in section 2 (before presenting the co2 results).

    - At the first measurements Guryong Tunnel, the distribution of CO2 is one-dimensional, simpler than the 2-D distribution in Bongcheon Intersecrtion, so we thought the two-point method would be enough. However, doing so resulted in significant increase of RMSE. We also applied the multi-point regression for the Guryong Tunnel to compare the results from the two methods.

    - We clarified the changes in the correction methods, along with the reasons for them, in Section 2.3. The previous Section 4.1 has been moved to Section 2.

- Section 2.3: This section would really benefit from a map, similar to what was provided in section 2.2

    - A figure (see below) was newly included in the revision.

[Figure]

Figure 1 Aerial photograph of Guryong tunnel and the location of placed HT-2000 sensors. Map data © Airbus 2025. The tick marks indicate the location of the sensors placed; 0 indicates the entrance to the tunnel.

- Section 3.1: Please list the coefficient values for the multiple linear regression, either in the main text or the SI. It's not clear how much of the correction is currently coming from the meteorological variables. After reading the whole article I see that Section 4.1.3 has some information on this. See earlier note about ordering. I still think it would be useful to list the coefficient values.

- The coefficient values are now included in the Supplement.

- I think Section 4.1 belongs in the methods section, not the discussion section. Several of my questions were answered when reading this section, and I would have liked to see calibration details before looking at the results.

  - We placed the previous section 4.1 inside section 2.

**Technical corrections**

- L30: This sentence seems a bit out of place, since the authors did not conduct eddy covariance measurements. Instead, I think the authors should motivate by referencing other studies that use low-cost sensor networks to monitor emissions.

  - The following words were deleted:

    "Ground-level $CO_2$ concentrations can be converted into atmospheric fluxes using flow measurement techniques such as eddy covariance (Burba et al., 2013; Vardag and Maiwald, 2023). Therefore, measuring ground-level $CO_2$ enables more accurate $CO_2$ flux estimation."

- L59: "The LI-840a boasts an accuracy better than 1.5% of the reading value and an RMS noise level below 1 ppm." I think this sentence should be stated the same way as the sentence describing the HT-2000 sensor to make for an easier comparison. Something like: "according to the manufacturer, its accuracy is within +/- XX ppm or +/- YY% of the reading." It's not clear if the RMS noise level of < 1ppm stated here is the same metric as the +/- 70ppm stated for the HT-2000.

  - We changed the sentence as follows:

    "According to the manufacturer, LI-840a has an accuracy better than 1.5% of the reading value and RMS noise level below 1 ppm. In addition, from our experience, the sensor's precision can be calibrated to significantly surpass the manufacturer's specifications, often yielding fluctuations of approximately 0.1ppm. Despite its superior performance, the high cost (>3000 USD) of the LI-840a unit limits the number of units that can be deployed simultaneously.

- L60-61: This claim seems unsupported by the work in this manuscript. Is there another study you can reference to support the 0.1% accuracy claim?

  - In general, LI-COR is popularly used for background measurements with precision better than 0.1 ppm. We have not independently checked it, but we assumed other labs also have achieved similar precision results. Considering the uncertainty of the low-cost sensors (~ a few ppm), even if the accuracy of LICOR is significantly larger than 0.1ppm, it would not make significant change in our conclusions.

- Fig 1: I only see 19 blue dots. Where is the 20th sensor located?

  - The sensor placed in one spot ended up having problems for the both measurements (in the Bongcheon Intersection), so it is not depicted in the map. The caption was changed to convey this information.

- L76: This sentence doesn't make sense: "The LI-840a was placed at a single location near the centre of the intersection using multiple-point linear regression." Perhaps this paragraph is out of order?

  - Sentence changed to read more smoothly:

    "We then corrected the 20 HT-2000 meters with multiple-point linear regression; the data for regression was obtained by placing all HT-2000 sensors and the LI-840a sensor at the same place, near the centre of the intersection."

- L90: Clarify that the covariates in the multiple linear regression are the fields measured by the HT-2000 sensor.

  - Description added as follows:

    "In this experiment, $y_i$ corresponds to the $CO_2$ concentration measured at time $i$ using the LI-840a analyser, while, $x_{i1}$, $x_{i2}$, and $x_{i3}$ represent the $CO_2$ concentration, temperature, and humidity (measured by each HT-2000), respectively."

- L142: Please state which interpolation method you are using with "*scatteredinterpolant.*" It's also worth noting that this method does not leverage

any information about, e.g., atmospheric transport or the presence of buildings that would block the flow of CO2 between potential sources and the sensor locations.

- We have used the default interpolation method which uses Delaunay triangulation to divide the plane into triangles, then linearly interpolates within the triangle. In results session, we mentioned the issue:

  "The corrected $CO_2$ concentrations obtained from each HT-2000 were then interpolated temporally for each HT-2000 sensor, then interpolated spatially using the *scatteredinterpolant* function in MATLAB which uses Delaunay triangulation then performs linear interpolation on each of the triangles on default settings."

- We believe we may sidestep the limitation (of not taking into advantage the urban landscapes / air transport) by adding more sensors. It is now mentioned in the Section 4.1 as follows:

  "Our methods do not incorporate any information about the urban landscape, such as building locations or shapes, nor do they rely on air-transport data. They can also be integrated with atmospheric modeling by replacing the spatial interpolation step with a suitable modeling approach. Because the sensors are low-cost, their numbers can be increased substantially, enabling much more comprehensive monitoring."

- Figure 4: It's unclear what the "distance from the entrance" means. Is this being measured into the tunnel, with 0m at the entrance? If so, at what distance is the exit point? It would be helpful to mark this distance on the figure. Also see earlier note about providing a map of the tunnel experiment. How many sensors were there and how far apart were they within the tunnel?

  - It represents the distance from the tunnel entrance. The map (currently figure 2) has been provided, with tick marks indicating the sensor locations.

- Figure 4: Needs some discussion of how the point data was spatial interpolated. Is it the same method as the intersection case study?

- Since the tunnel is linear, we have linearly interpolated each sensor in time first, then linearly interpolated spatially, using all sensors. Former figure 4 (currently figure 9) caption has been changed as follows:

  - "Figure 9. $CO_2$ values measured (corrected) at the Guryong Tunnel, on July 25 (a) and November 21 (b, segment) in 2024. Data was temporally resampled from 10 s between each data points to 1 s using linear interpolation, then spatially interpolated for each second using 1-dimensional linear interpolation. The y-axis is measured from the tunnel entrance, and the 1180m mark represents the exit of the tunnel."

- Fig 10: please state where the traffic data are coming from

  - The traffic data were obtained from Seoul Transport Operation and Information Service (https://topis.seoul.go.kr/refRoom/openRefRoom_1.do).

  - We added the data source in the Fig. 11 caption (note that the figure numbers were changed in the revision).

---

## Author Comment (AC2)

*We would like to thank the two anonymous reviewers and the Editor for their careful reviews of our manuscript. Below we present our point-by-point responses to all of the comments. The original comments from the reviewers are shown in black, while our responses are presented in blue.*

**Reviewer #2**

The authors aim to address the spatial heterogeneity and temporal variability of urban $CO_2$ by applying a multiple-point linear regression to low-cost sensor measurements. They present two experiments—one at a high-traffic intersection (Bongcheon) for several months, and another in a tunnel over two days—and claim that high-resolution $CO_2$ maps can be generated with this approach.

I remain skeptical about the practicality of this linear-regression-based interpolation method. A linear model ignores the geometric constraints inherent to urban environments. Moreover, the paper provides insufficient detail on how the regression coefficients are learned, leaving open the possibility of overfitting. Finally, the authors do not discuss the spatial and temporal variability of those coefficients, which is essential for assessing the method's real-world applicability.

**General comments**

- **Linear interpolation lacks physical realism.** A purely linear model does not incorporate the geometric constraints that govern $CO_2$ transport in cities. This shortcoming is evident in the presented concentration maps, which show little correlation with major roadways.

  - We focused on the potential of using low-cost sensors to improve spatial resolution. The mechanical transport modeling combined with measurements is not the main focus of this article. Although our number of sensors was not sufficient to fully capture 3-D $CO_2$ movement, our 2-D measurements clearly show elevated $CO_2$ concentrations at Bongcheon Intersection, where vehicles stop or move slowly due to traffic signals (see Video S1~S4). We also found

that $CO_2$ concentrations were strongly correlated with vehicle counts in Guryong Tunnel (Figure 11).

- We note the limitations and potential of the mapping methods in Section 4.1 by adding the following text:

"Our methods do not incorporate any information about the urban landscape, such as building locations or shapes, nor do they rely on air-transport data. They can also be integrated with atmospheric modeling by replacing the spatial interpolation step with a suitable modeling approach. Because the sensors are low-cost, their numbers can be increased substantially, enabling much more comprehensive monitoring."

-

- Also, the manuscript does not explain that each spatial coordinate would require its own set of regression coefficients and time-alignment parameters. A discussion of the spatio-temporal variability of these coefficients is necessary if the method is to be useful in practice.

  - We made correction equations for each low-cost sensors and assumed no change of coefficients during measurement. We analyzed the concentration only up to 24 h due to limited time of battery capacity. However, we take advantage of the very low cost of 85 USD compared to other low-cost sensor sets (1000s USD).

  - During measurements at the Guryong Tunnel, we found that the sensors remained stable for 24 hours. We expect the sensors to remain stable for a longer period, but the HT-2000's battery life is insufficient to operate them for more than 48 hours. We note this limitation in Section 2.1.

- **Training procedure for the multiple-point linear regression is under-described.** The authors do not specify how the data are split into training and test sets, nor whether any cross-validation is employed to guard against over-fitting. Without this information, the reported RMSE metric cannot be trusted.

- There is no "learning" period since it is a mathematical technique of finding the best fit line that minimizes residual errors (no LLM involved). To clarify how the coefficients for each sensor were obtained, we added more explanation and specified the procedure in detail.

- Figure S2 has been newly added to show the time series used for when low-cost sensors and LICOR were co-located for correction (i), (in transit (ii)), and HT-2000 sensors were deployed for measuring data to create spatial maps (iii).

- **Calibration, correction, and interpolation are conflated.** At times it is unclear whether the authors refer to sensor calibration, correction, or spatial interpolation. These three steps serve distinct purposes and should be clearly separated.
- We reviewed the relevant sentences and improved the clarification. To remove confusion, we used the "calibration" only for the LI-COR, and now we newly used "correction" for any methematical works for the low-cost sensors after measurements which includes calculation of coefficients for each sensor and change the raw data to fit the LICOR measurement values.

**Specific comments**

- **l. 17** – Clarify the local CO2 sources in Seoul (e.g. transportation, heating). When citing the city's 9 % share of national electricity consumption, indicate whether the associated emissions stem from power plants within Seoul or elsewhere; otherwise the statement lacks context.

  - Seoul directly emitted 18,565 ktonCO$_2$ in 2022; The two most important direct sources of CO$_2$ for it are buildings (commercial/residential) and road traffic; thoseare responsible for nearly 80% of all CO$_2$ emissions within Seoul. In addition to that, Seoul was indirectly responsible for 19,727 ktonCO$_2$ in 2022, where about 82% of which is from power generation. (Seoul Carbon Neutrality Support Center, 2022) The power plants are located outside Seoul, so it is indirect emission.

  - Line 17 was changed as follows:

    "Seoul is a megacity, home to approximately 20% of the country's population despite occupying only 0.6% of the total land area (Seoul Metropolitan

Government, 2024; Korean Ministry of Culture, Sports and Tourism, 2024). Seoul directly emitted 18,565 ktonCO$_2$ in 2022; The two most important direct sources of CO$_2$ for it are buildings (commercial/residential) and road traffic; those 2 are responsible for nearly 80% of all CO$_2$ emissions within Seoul. In addition to that, Seoul was indirectly responsible for 19,727 ktonCO$_2$ in 2022, where about 82% of which is from power generation. (Seoul Carbon Neutrality Support Center, 2022)"

- **l. 21** – Explain why a high-resolution CO$_2$ map would aid mitigation strategies. A brief illustrative example would help readers understand the use case.

  - A CO2 measurement of this detail can reveal different point sources within the urban landscape, usually treated as a single plane source of CO2 due to insufficient resolution. As well, it could reveal different phenomena that increases or decreases local CO2, such as the effect of idling vehicles.
    We now newly added the following words:

    "for example, accurate high-resolution monitoring of urban CO$_2$ may reveal new point sources within the urban landscape, usually treated as a single plane source. As well, it may reveal important phenomena that influences local CO$_2$ level. "

- **l. 25** – Define "top-down" and "bottom-up" approaches.

  - Top-down estimation of CO$_2$ inventories starts from measurements of CO$_2$ concentrations, then tries to convert it into CO$_2$ flux. In contrast, Bottom-up estimation of CO$_2$ inventories start from the compilation of CO$_2$ emitters and calculates the total CO$_2$ emission from them directly. We now newly added the following sentences:

    "This variability complicates accurate estimation of CO$_2$ fluxes at small scales using a top-down approach, which begins with concentration of CO$_2$ and calculates flux from it, making urban flux calculations predominantly reliant on a bottom-up methods, which relies on statistic data to compile CO$_2$ inventories. "

- **l. 30** – How ground-level CO2 measurements can improve eddy-covariance flux estimates ?

  - We have not performed any eddy covariance measurement. The following words were deleted:

    "Ground-level $CO_2$ concentrations can be converted into atmospheric fluxes using flow measurement techniques such as eddy covariance (Burba et al., 2013; Vardag and Maiwald, 2023). Therefore, measuring ground-level $CO_2$ enables more accurate $CO_2$ flux estimation". Nonetheless, we believe that more detailed CO2 measurement will enable calculating more accurate CO2 flux by reducing errors in measuring CO2 concentration.

  **l. 35**– Both sensors are using NDIR, maybe specify the exact sensor models used.

    - HT-2000 sensors use SenseAir S8 sensor as already was written in the submitted manuscript. We now newly added the following sentences in section 2.1: "the LI-840a operates as a closed-path system using a custom-made optical bench."

- **l. 43** – State the main contribution explicitly: the development of a spatial-interpolation method for CO2 using low-cost sensors.

  - The main contributions are twofold: first, we developed and validated a method for calibrating a low-cost NDIR sensor; second, we demonstrated that these low-cost sensors can be effectively applied to high-spatial-resolution mapping of $CO_2$ concentrations at specific locations in Seoul.

  - The final paragraph in Introduction now reads as such:

    "This study presents an approach using a very low-cost, pre-assembled $CO_2$ monitoring kit (85 USD; including nondispersive infrared (NDIR) $CO_2$ sensor, relative humidity sensor, and temperature sensor) to enable high-resolution $CO_2$ measurements in urban environments. We also present 2D spatial movies of ambient $CO_2$ levels measured in urban Seoul and within a tunnel

in the city, and visualizations made using the resulting data. Time-series data were corrected post hoc using a high-precision NDIR sensor (LI-840a), and we discuss methodologies for implementing effective correction schemes."

- **l. 45** – Distinguish clearly among calibration, correction (post-processing of raw measurements to remove spikes, drift, etc.), and interpolation/estimation (predicting CO2 at unmeasured locations, possibly after time-lag alignment).

  - To remove confusion, we used the "calibration" only for the LI-COR, and now we newly used "correction" for any methematical works for the low-cost sensors after measurements which includes calculation of coefficients for each sensor and change the raw data to fit the LICOR measurement values. We clarified this by revising the wording as follows:

  - - In Section 3.2: "The time-averaged, corrected $CO_2$ values from the meters ranged from 348 to 482 ppm during the 2022 session, and from 457 to 512 ppm during the 2023 session. The **corrected** $CO_2$ concentrations obtained from each HT-2000 were then **interpolated** temporally for each HT-2000 sensor, then **interpolated** spatially using the *scatteredinterpolant* function in MATLAB which uses Delaunay triangulation then performs linear interpolation on each of the triangles on default settings."

- In Section 2.3.3: "To evaluate whether the multi-point linear regression method produces better results, we compared it with a flat-shift method. For the Bongcheon Intersection measurements, the measurement error was assumed to be constant. In the flat-shift method, each HT-2000 device was treated as having a fixed bias relative to the actual $CO_2$ concentration, and this bias was used for the correction. During correction, the data from individual HT-2000 unit were averaged and compared with the average of LI-840a data. The HT-2000 data were then adjusted to match this average. For comparison, a multi-point linear regression method was also used to correct the HT-2000 devices. Both methods significantly reduced the RMSE values, with the multi-point linear regression method reducing the median RMSE by about 0.7 ppm, compared to the flat shift correction method (**Error! Reference source not found.**). This suggests that most of the errors stem from a constant offset from the

true value. However, because multi-point linear regression also showed a meaningful (~20%) reduction in the median RMSE, it is reasonable to conclude that scaling errors are also present in low-cost NDIR sensors."

- **l. 60** – Provide the cost ratio between the two sensor types to understand the trade-off between price and accuracy.

  - Approximate cost of the both sensors are now provided in the manuscript. (85 vs >3000 USD)

- **l. 64** – Consider reorganizing the methods section into three subsections: "Instrument description," "Site description," and "Interpolation method."

  - Section 2 modified into three subsections; 2.1 Instrument description, 2.2 Site description and measurement procedure, 2.3 Data correction method.

- **l. 80** – Explain the nature of the reported time delays: are they instrument-specific latencies, or delays relative to the target interpolation location? Discuss how the method would handle interpolation at a location lacking any ground-truth measurement, and describe the optimization technique used for time-lag correction.

  - Those are instrument specific delays. The technique outlined in paper would still work without ground truth measurements, but it would only measure differences from an indeterminate value.

  - Sentence changed in section 2.2 as follows:

    "We have tried various lengths of smoothing window using a FOR loop for both correction periods and found out that 137 seconds yielded the least RMSE difference between the LI-COR and HT-2000 instruments. The time delay for each HT-200 was also calculated algorithmically, by finding the "best match" between the LI-COR and HT-200."

- **l. 84–95** – Indicate which portion of the dataset is used to learn the regression coefficients and whether any cross-validation scheme is applied to prevent over-fitting.

- The figure S2 shows the time sections of data used for co-located measurement for obtaining data for correction of HT-2000 (i), moving HT-2000 sensors to its assigned location (ii), HT-2000 measurement in its assigned location (iii). Since no neural nets were used, no cross-validation schemes were used.

- **l. 90** – Justify the inclusion of temperature and humidity as covariates in the interpolation model.

  - In the early stage of the work (measurements at Bongcheon Intersection), we included both as independent variables. However, the contribution from it was not significant, and thus in the late stage in the measurements at Guryong Tunnel it was not included in the regression. We do not think it is useful to include temperature or humidity in the regression, since the contributions from them are small. To clarify the change, section 2.3.5 now contains those sentences:

    "Absolute humidity and temperature were included as factors in the multi-point linear regression during Bongcheon Intersection measurements, because we suspected residual interference in the low-cost NDIR sensor. To evaluate the impact of humidity and/or temperature in the regression, we performed a multi-point linear regression with different combinations of factors. Including absolute humidity or temperature did not significantly change the median RMSE values, as shown in **Error! Reference source not found.** (ranging from 5.10 to 5.21 ppm in 2022 and from 4.09 to 4.71 ppm in 2023). Thus, it was not included as factors during Guryong Tunnel measurements."

- **l. 114** – Clarify whether the displayed equation is for to sensor calibration or to the spatial interpolation itself.

  - The equation is for correcting the sensor post-measurement. The 2.3.2 section is rewritten to include this sentence:

    "It was used to apply post-measurement corrections to the HT-2000 sensors."

- **l. 127** – The RMSE appears to compare CO2 values at two distinct sites simultaneously. One would expect a comparison between the interpolated value and the ground-truth.

  - The RMSE calculation was only applied to the data obtained from co-located measurements (i.e., measured with LI-840a and HT-2000 at the same location). A sentence added in Section 3.1:

    "For the Bongcheon Intersection measurement, we first measured the $CO_2$ concentration simultaneously with LI-840 and HT-2000 on the same spot near the center of the intersection to provide information for correction. "

- **l. 135** – Confirm whether the low-cost sensor time-series are temporally aligned before regression.

  - They were aligned temporally then linear regression was performed to obtain the coefficients. Figure 7 caption was modified as follows:
    "Figure 7. CO2 concentrations measured in Bongcheon Intersection, before and after correction. a) and b) Plots of CO2 values measured by HT-2000" (grayscale) and LI-840a (red), before any correction was applied. c) and d) The LI-840a values smoothed with a 137-second window. Note that the HT-2000 sensor values closely follow the smoothed LI-840 values, though with fixed offsets. e) and f) Results after linear correction of the HT-2000 meters."

- **l. 142** – If the interpolation described here is identical to the multiple-point linear regression introduced at l. 94, state this explicitly and reference the MATLAB implementation in the "Interpolation method" subsection.

  - We think this reviewer has misunderstood. *scatteredinterpolant* was used to spatially interpolate the $CO_2$ concentrations, but two processes were performed beforehand: first, we corrected all HT-2000 sensors (this is the procedure detailed in L94), then linearly interpolated the HT-2000 sensor data to increase the data resolution from 3s to 1s.   Sentences in 3.2 was rewritten to clarify this issue:
    "The corrected $CO_2$ concentrations obtained from each HT-2000 were then

interpolated temporally for each HT-2000 sensor, then interpolated spatially using the *scatteredinterpolant* function in MATLAB which uses Delaunay triangulation then performs linear interpolation on each of the triangles on default settings."

- **l. 143** – Expand the discussion of results: why is linear spatial interpolation justified in a complex urban environment? Address the apparent lack of correlation between major roads and elevated CO2 levels.

  - The justification is detailed in section 4.2:

    "Videos showing the rate of $CO_2$ concentration change (Supplement S2 and S4) revealed quasi-periodic variations at the junction. The four sensors positioned at the junction were located on traffic islands, where the primary source of $CO_2$ is vehicular traffic. Specifically, vehicles idling at a red light contribute to a concentration build-up at fixed locations, whereas vehicles in the right-turn lane do not have the same effect. Therefore, it is reasonable to associate these $CO_2$ changes with traffic signal cycles. A similar build-up was also observed in the northeastern corner of the 2023 video (Supplement S4), where the responsible sensor was positioned near a traffic crossing."

- **l. 153** – Specify whether the tunnel is two-way.

  - The tunnel is made of two one-way tunnels. We only measured from one side. It was newly described in the site description in section 2.

- **l. 168–176** – Separate statements about calibration from those about data interpolation to avoid confusion.

  - Former Section 4.1.1 (now 2.3.3) mentions about correcting HT-2000 sensors. The mentioned section (L168-176, now the first paragraph of 2.3.3) did not contain discussions about data interpolations. All data interpolations (temporal and spatial) were done linearly to avoid introducing mathematical artifacts.

- **l. 193–205** – The reported decrease in error from the "time-lag corrections" seem insignificant compared with the reading error (HT-2000, LI-870A). The observed

variability (> 10 s) may stem from the optimization algorithm rather than true physical lag.

- For the 2022 Bongcheon Intersection measurement, the time delay was significant, and it resulted in a large median RMSE increase of 2 ppm. Thus, we think it is important to correct time delays between the CO2 kits.

- **l. 194** – Describe the algorithmic procedure used to estimate the time lag.

  - The time-lag was calculated by shifting the measurements in various increments then calculating the variance of the difference between the LI-COR measurement vs. HT-2000 measurement. The Matlab code was newly described in the Supplement.

- **l. 207–211** – As above, the error analysis does not appear meaningful when it is of the order of the sensor's reading error.

  - We think it is a useful information: The error analysis showed that humidity or temperature has significantly less effect on the measured CO2 of HT-2000 kits, and the error of internal NDIR sensor was far more important.

- **l. 215–225** – I disagree with the notion of a universal set of regression coefficients for all locations. Coefficients inevitably depend on local geography and prevailing atmospheric conditions.

  - Lines 215-225 currently corresponds to section 2.3.6. Like the reviewer has pointed out, we also have determined that having one set of regression coefficients results in a large increase in error (as written in the section).

- **l. 262** – The concluding claim that the approach enables large-scale, high-resolution monitoring is premature. The manuscript does not examine the spatial or temporal stability of the regression coefficients, which is essential to validate the method.

  - The HT-2000 devices can operate for only about 48 hours due to battery limitations, but their runtime could be extended if continuous power is supplied. This has now been noted in Section 2.1.